# Influence of Codiagnosis of Chronic Fatigue Syndrome and Habitual Physical Exercise on the Psychological Status and Quality of Life of Patients with Fibromyalgia

**DOI:** 10.3390/jcm11195735

**Published:** 2022-09-28

**Authors:** María Dolores Hinchado, Eduardo Otero, María del Carmen Navarro, Leticia Martín-Cordero, Isabel Gálvez, Eduardo Ortega

**Affiliations:** 1Immunophyisiology Research Group, Instituto Universitario de Investigación Biosanitaria de Extremadura (INUBE), Av. de Elvas s/n, 06080 Badajoz, Spain; 2Immunophysiology Research Group, Physiology Department, Faculty of Sciences, University of Extremadura, 06071 Badajoz, Spain; 3Immunophysiology Research Group, Nursing Department, University Center of Plasencia, University of Extremadura, 10600 Plasencia, Spain; 4Immunophysiology Research Group, Nursing Department, Faculty of Medicine and Health Sciences, University of Extremadura, 06071 Badajoz, Spain

**Keywords:** fibromyalgia, chronic fatigue syndrome, exercise, stress, COVID-19

## Abstract

Fibromyalgia (FM) and Chronic Fatigue Syndrome (CFS) are two diseases that are frequently codiagnosed and present many similarities, such as poor tolerance to physical exercise. Although exercise is recommended in their daily routine to improve quality of life, little is known about how CFS codiagnosis affects that. Using scientifically validated questionnaires, we evaluated the psychological state and quality of life of patients with FM (n = 70) and how habitual physical exercise (HPE) reported by patients with only FM (FM-only n = 38) or codiagnosed with CFS (FM + CFS, n = 32) influences those aspects. An age-matched reference group of “healthy” women without FM (RG, n = 70) was used. The FM-only group presented a worse psychological state and quality of life compared to RG, with no influence of CFS codiagnosis. The patients of the FM-only and FM + CFS groups who perform HPE presented better levels of stress and state anxiety, but with no differences between them. Depression and trait anxiety improved only in women with just FM. CFS codiagnosis does not worsen the psychological and quality of life impairment of FM patients and does not have a great influence on the positive effect of HPE.

## 1. Introduction

Approximately 6.3% of the world population suffers from fibromyalgia (FM), being more frequent in women, according to the World Fibromyalgia Association, and although it is a chronic disease with no fully established etiology, moderate physical exercise is the main proven, validated non-pharmacological therapy for the management of FM according to the EULAR [1]. The problem lies in the fact that the pain and fatigue felt by people with FM often hamper participation in regular daily activities and even more so, in sporting activities. Despite that, physical exercise is the most recommended adjuvant in the treatment for the improvement of the predominant symptom: pain [2]. In addition, people with FM who have a better physical shape also have a reduced manifestation of this condition [3]. Although a positive relationship between habitual physical exercise (HPE) and quality of life in FM has been clearly described [4,5,6,7,8], one of the great challenges of FM is the frequent association of this syndrome with different pathologies, such as Chronic Fatigue Syndrome (CFS). CFS is characterized by disabling fatigue lasting more than 3 months, which further worsens the course and development of the disease and makes it even more difficult to acquire routines that include exercise. For people with CFS, exercise also seems to significantly improve health status, physical fitness, and quality of life, thus having a direct impact in fatigue, but literature is still scarce in this regard. People with CFS tend to have an aversion to exercise to avoid symptom aggravation. Because of this, they do not (attempt to) engage in physical activities and become completely sedentary [9]. Therefore, little is known about the benefits of daily exercise practice when these two syndromes are comorbid, which occurs in 20–81% of FM patients [10,11]. In fact, several authors already pointed out that it is important to consider the comorbidity of both pathologies when recommending physical exercise, since they found differences in the perception of the intensity of effort and a decrease in systolic blood pressure [12]. Moreover, FM + CFS patients also had moderately lower total peripheral vascular resistance and experienced greater musculoskeletal pain during exercise than FM-only patients. In women with only FM, aerobic exercise improves the levels of anxious and depressive symptoms that develop in numerous patients with this pathology [6]. Benefits of exercise therapy have also been found in patients diagnosed with only CFS, especially in terms of fatigue, without evidence that it may worsen the course of the disease. However, more studies are needed to examine the influence of CFS comorbidity on the effects of HPE in FM patients, particularly concerning quality of life, anxiety, and depression [13]. Therefore, in this research, it seems plausible to question how the codiagnosis of CFS has a perceived influence on the effects of reported regular physical exercise on psychological well-being and quality of life in patients with FM.

Previous studies in our laboratory have shown that pain, fatigue, and decreased quality of life of FM patients are due, at least in part, to a neuroimmunoendocrine dysregulation in stress and inflammation biomarkers. This dysregulation is improved with the performance of HPE [5,7,8,14]. Nevertheless, and taking into account that fatigue is the main symptom in CFS patients, it is plausible to hypothesize that CFS codiagnosis in FM patients could affect their quality of life and psychological state, as well as the effect of habitual physical activity on these aspects. In this context, and before delving into neuroimmunoendocrine biomarkers, the objective of this study was to ascertain how codiagnosis of CFS and the performance of HPE affects psychological health and perceived quality of life in patients with FM, all of this in an easy and non-invasive way, which is part of a broader investigation. This will allow a deeper insight into the influence of HPE (the main non-pharmacological therapy available in this pathology) reported by the patients, on psychological aspects and quality of life in patients with FM. Understanding of these aspects will contribute to clarifying other psycho-neuroimmunoendocrine axis mechanisms and explore strategies that improve the course of both diseases.

## 2. Materials and Methods

### 2.1. Participants

Extremadura is an autonomous community of Spain with approximately 1,000,000 inhabitants, with a very homogeneous population in terms of lifestyle. The majority of the population is assisted by the Spanish National Health System, in which patient associations play a relevant role. Extremadura is also a reference region in Spain in health research [15,16], and particularly in patients with fibromyalgia and in the effects of exercise internationally.

The study was carried out in 70 women diagnosed with FM (Total FM group), all of them aged between 40 and 65 years, and belonging to the FM associations of Extremadura. A total of 38 of these women had a diagnosis of only FM (FM-only group), and the remaining 32 also had a previous codiagnosis of CFS (FM + CFS group). In total, 70 women of the same age range were used as a reference group of “healthy” women, not diagnosed with FM, CFS, or any other inflammatory or rheumatic pathology (RG), as well as pathologies affecting depression, anxiety, and/or pain. Figure 1 shows the flow charts of participants in the study.

It is important to highlight, as a strength of this study, that among the volunteers belonging to the FM associations of Extremadura, all of those who met the inclusion criteria were selected: (a) diagnosis of CFS and/or FM by rheumatologists or internal medicine professionals according to ACR diagnostic criteria for FM patients [17], and Fukuda and co-workers criteria for CFS patients [18], (b) aged between 40 and 65 years, (c) not having a diagnosis of depression, and (d) not suffering from multiple chemical sensitivity. Table 1 shows the main characteristics of that participants: anthropometrics data, employment status, and common comorbidities. All participants were Caucasian women and had been diagnosed with FM (with or without a previous diagnosis of CFS) for more than two years. There were no significant differences in age and BMI between the groups. With respect to professional work activity and the most common comorbidities of participants, no differences were found between FM-only and FM + CFS groups, but they presented significant differences with respect to the RG group, particularly in white collar workers and retired people as well as women diagnosed with osteoarthritis. Medication history was very diverse in each patient, but all of them were prescribed, at time of the present study, with different types of anti-inflammatory and analgesic drugs (e.g., ibuprofen, dexketoprofen, paracetamol, tramadol). All participants of experimental and control groups diagnosed with hypothyroidism were treated with levothyroxine.

Written informed consent was also requested from all participants before participating in the study. The research had been previously approved by the Bioethics Committee of the University of Extremadura by the Directives of the Council of Europe and the Declaration of Helsinki (registration number 13/2020). This study was registered with ClinicalTrials.gov (identifier: NCT05323838: available on the website).

### 2.2. Instruments

To evaluate the perceived quality of life in our experimental groups, we used the following scientifically validated questionnaires.

The Beck Depression Inventory (BDI) was used to determine the presence of signs of depression during the last week, including the day of the test. Higher scores are related to greater signs of depression. According to the final score, perceived depression can be classified as: mild (between 10–19), moderate (between 20–30) or severe (>30) [19]. Spanish version of Sanz et al. [20] was used.

The Perceived Stress Scale (PSS) is a self-report instrument that evaluates the level of perceived stress during the last month, consisting of 14 items with a five-point Likert scale response format (0 = never, 1 = almost never, 2 = sometimes, 3 = fairly often, 4 = very often). The total score of the PSS is obtained by reversing the scores of items 4, 5, 6, 7, 9, 10 and 13 (as follows: 0 = 4, 1 = 3, 2 = 2, 3 = 1 and 4 = 0) and then adding up the scores of the 14 items. A higher total score corresponds to a higher level of perceived stress [21]. The Spanish version of Remor [22] was used.

The State-Trait Anxiety Inventory (STAI) is a 40-item self-report questionnaire designed to assess two independent concepts of anxiety: state anxiety (transient emotional condition) and trait anxiety (relatively stable characteristic of anxiety propensity). Each subscale comprises a total of 20 items in a four-point Likert response system according to intensity (0 = not at all/almost never, 1 = somewhat/sometimes, 2 = moderately so/often, 3 = very much so/almost always). The total score on each of the subscales ranges from 0 to 60 points [23]. Higher scores indicate a higher state of anxiety. Spanish version by Buela-Casal & Guillén-Riquelme [24] was used in the present study.

The Brief Pain Inventory (BPI) is a self-administered questionnaire that was originally designed to assess cancer pain [25]. It is now also used as a generic pain questionnaire for other chronic pain conditions [26]. It consists of two basic magnitudes: intensity, and interference of pain with the patient’s activities of daily living, both scored on scales from 0 “no pain” to 10 “the worst pain”. Higher scores are directly correlated to a higher perception of pain. The Spanish version of the BPI has proven to be valid for measuring the intensity of pain and its impact on activities of daily living under routine clinical practice conditions [27].

The Brief Fatigue Inventory (BFI) is a questionnaire that assesses the level of fatigue and its impact on the activities of daily living of the subjects in the last 24 h [28]. The first three items evaluate the subject according to their perceived fatigue on a scale from 0 “no fatigue” to 10 “the worst fatigue”, and the remaining items evaluate the interference of fatigue in different aspects of the subjects’ lives (general activity, mood, walking ability, work, relations with other people, and enjoyment of life) on a scale of 0 “does not interfere” to 10 “completely interferes”. The higher the score of both items, the greater the perceived fatigue. The Spanish version of Valenzuela et al. [29] was used.

The main objective of the Healthy Lifestyle and Personal Control Questionnaire (HLPCQ) is to detect and quantify lifestyle patterns that reflect the health improvement, as evidenced by stress levels and the internal health locus of control [30]. The following sections were evaluated: choosing a healthy diet, avoiding a harmful diet, daily routine, organized physical exercise, and social and mental balance. With this instrument, it was possible to conclude whether the individual was able to maintain adequate life control.

In 1994, Burckhardt et al. [31] developed a specific tool to measure the impact of FM on the functional capacity and quality of life of people who present this pathology: the Fibromyalgia Impact Questionnaire (FIQ). The FIQ assesses the impact of FM on physical functioning, the ability to perform regular work and, in the case of having a paid job, the degree to which FM has affected this activity, as well as subjective items closely related to the FM clinical profile (pain, fatigue, tiredness, and stiffness) and emotional state (anxiety and depression). Spanish version of the questionnaire was used. To obtain the total score, the different items were normalized, as a result the total score oscillated between 0–80 [32]. A higher score indicates a negative impact of FM on the patient’s health.

Lastly, and given the pandemic era we are living in, we decided to include questionnaires that evaluated fear and anxiety towards Coronavirus Disease 2019 (COVID-19): the Coronavirus Anxiety Scale (CAS) and the Fear of COVID-19 Scale (FCV-19S).

The CAS is a brief mental health assessment that can be used to identify cases of dysfunctional anxiety related to COVID-19 [33]. The items measure physiological symptoms that are awakened by information and thoughts related to the coronavirus using a 5-point time-anchored scale (0 = not at all, to 4 = almost every day during the last 2 weeks). The higher the score, the greater the perceived anxiety related to COVID-19. The Spanish version of Caycho-Rodríguez et al. [34] was used.

The FCV-19S is used to identify people with high levels of fear of COVID-19, and perform early psychological interventions [35]. It is a 7-item scale that is scored using a 5-point Likert scale, ranging from 1 (totally disagree) to 5 (totally agree). Total scores can range from 7 to 35, with higher scores indicating greater fear related to COVID-19. The Spanish version of Sánchez-Teruel and Robles Bello [36] was used.

### 2.3. Procedure

Quality of life and psychological status of patients with FM, with or without codiagnosis of CFS, and with self-reported HPE, were evaluated comparatively, all of this in relation to the RG of the same age range. All participants filled a questionnaire answering about performance of physical activity in the last 3 months. This questionnaire asked participants about intensity, frequency, and type of exercise, as well as their adherence to the program. All types of supervised programs of regular physical exercise for FM patients performed in the 3 months prior to completing the questionnaires, for a minimum of two hours a week on alternate days, were considered as HPE (Table 2). Daily activities could not be evaluated, but participants did not refer to take a walk or bike ride to the job. No differences were found in the professional work activities between the experimental groups (Table 1). All questionnaires were filled by all participants under supervision in December 2020.

### 2.4. Analysis of Data

Values are expressed as mean ± standard error of the mean (SEM). The variables were normally distributed tested by the Kolmogorov–Smirnov normality test. Student’s *t*-test was used for comparisons between groups (paired samples). Chi-square independence test and z-test for independent proportions with Bonferoni corrections was used for comparisons between percentages in Table 1 and Table 2. Minimum significance level was set at *p <* 0.05. Statistical analysis was performed with the SPSS^®^ Statistics v.27.0 package.

## 3. Results

### 3.1. Psychological State and Quality of Life

Figure 2 shows the psychological state and quality of life of our group of women diagnosed with FM. FM patients showed worse values (*p <* 0.001) of depression (Figure 2a), stress (Figure 2b), anxiety (Figure 2c), pain (Figure 2d), fatigue (Figure 2e), impact of fibromyalgia (Figure 2f) and greater fear (Figure 2g) and anxiety towards COVID-19 (Figure 2h) compared to RG.

Once corroborated, as expected, the deterioration in psychological health and quality of life of our population, we wanted to know how the diagnosis of FM, with and without codiagnosed CFS, affects the performance of regular physical activity reported by the study volunteers (Table 2). Paradoxically, the group of patients with FM + CFS presented a higher percentage of women who reported performing HPE, even in the same order of magnitude as the group of healthy women. No differences were found between the experimental groups in the type of HPE reported by the participants. In addition, any correlation between participants reporting HPE and their daily professional work or common comorbidities could be determined.

### 3.2. Influence of Codiagnosis of CFS and Performance of HPE

Figure 3 shows the influence of the codiagnosis of CFS and the performance of HPE on the psychological state and mental health of women with FM. The results clearly show that codiagnosis of CFS does not affect (*p* > 0.05) levels of depression (Figure 3a), stress (Figure 3b), state anxiety (Figure 3c) or trait anxiety (Figure 3d). In both groups separately, worse values were manifested again in the evaluated parameters (*p <* 0.001) with respect to the RG. However, patients with FM (both in the presence and in the absence of codiagnosis of CFS) who report performing HPE present significantly (*p <* 0.05) better levels of depression, stress, and anxiety than sedentary ones, even reaching anxiety values that are very close to the RG.

When COVID-19-related anxiety was studied in a specific way (Figure 3e), the behavior followed the same pattern, but it was substantially exacerbated. The intensity of this particular anxiety might be the reason why physical activity did not improve it significantly, although a strong tendency to improvement is observed. When fear was assessed (Figure 3f), the pattern was similar, although in this case, both groups of women (FM-only and FM + CFS groups) who reported performing HPE had lower levels of fear (*p <* 0.05), even almost reaching the level of our RG.

Figure 4 shows the influence of the codiagnosis of CFS and the performance of HPE on quality of life of our group of patients with FM. The codiagnosis with CFS did not affect quality of life and personal control (Figure 4a), nor pain (Figure 4b) of the patients with FM, although a worsening of all the parameters was observed compared to our RG. Only when comparing the results obtained in the complete FM + CFS group, both sedentary and performing HPE, a statistically significant (*p <* 0.01) greater fatigue was observed compared to the complete group of patients diagnosed with only FM (statistical significance that does not appear in Figure 4c. Patients with FM (both in the presence and in the absence of codiagnosis of CFS) who reported performing HPE presented significantly higher levels of quality of life and personal control (*p <* 0.001) in the only FM group and *p <* 0.01 in the FM + CFS group) than sedentary women, reaching the values of our RG. However, no improvement in fatigue and pain levels was observed in women with FM who reported performing HPE neither in the presence nor in the absence of a codiagnosis of CFS. Finally, only in FM-only group, HPE seem to have a positive influence on the impact of fibromyalgia on daily life (*p <* 0.05) (Figure 4d).

Finally, Table 3 shows the items evaluated in the HLPCQ questionnaire, with their corresponding statistics.

No significant differences in any score corresponding to diet, organized exercise, and social and mental balance were found among participants from the three groups. However, participants reporting HPE improved dietary healthy choices and organized physical exercise scores in the three groups, and social and mental balance only in the FM + CFS group. Total score of HLPCQ improved in participants reporting HPE from the three groups.

## 4. Discussion

In recent years, our research group has evaluated different immunoneuroendocrine mechanisms underlying the pathophysiology of FM syndrome, clearly observing a neuroimmunoendocrine dysregulation between immune and stress responses that can be improved by physical exercise [5,7,8,37]. These studies have been carried out in patients belonging to different associations from the autonomous community of Extremadura (Spain). The population, due to its socio-sanitary characteristics, lifestyle, and follow-up by the public health system, has already been referred to in many studies as suitable for scientific studies on health and well-being [15,16]. Additionally, in terms of FM, this region is a reference throughout Europe due to the abundance of research, from different approaches, by different groups. However, a practical problem that arises is the need for diagnosis and therapeutic management in primary healthcare and psychological consultations in a simple manner, that is, without expensive clinical analyses and tests evaluating biomarkers of inflammation/stress and biomarkers associated with fatigue and pain mediators. In this context, it is also essential that professionals can address the differential (or similar) management of patients with FM and/or CFS as diagnosed by rheumatologists or internists either jointly or in separate diagnoses.

Although FM and CFS are two different diseases with different diagnostic criteria, there is a considerable overlap between the two as they share many clinical features. From a pathophysiological point of view, it is thought that both conditions are caused by the same alterations involving inflammatory, infectious and/or autoimmune components [38]. Contrary to that described in the work by Faro et al. [39], where the codiagnosis of FM in patients with CFS worsens clinical parameters, fatigue, and quality of life perception, in the present investigation we found more similarities than differences in these two alterations. Our results showed that the codiagnosis of CFS did not significantly affect the parameters evaluated in patients with FM, and that physical exercise, in these patients as reported in previous investigations from our group in FM patients, is an effective non-pharmacological therapy for the alleviation of some of the dysregulations found in these patients. In the present work, we had a homogeneous group of volunteers in which approximately half of the patients with FM had also been diagnosed with CFS. All of them, as expected, showed worse values in all the measured parameters regarding mental health and impact of FM on quality of life than the group of “healthy” volunteers of the same age range. Nevertheless, HLPCQ, a general questionnaire for evaluating quality of life but not specific for FM patients, did not showed differences between FM patients and the control group. This could reveal that this questionnaire may not be appropriate for evaluating the impact of FM in the quality of life of these patients, although it does reflect the beneficial effects of participation in regular physical exercise programs. Paradoxically, almost half of the volunteers with FM codiagnosed with CFS reported performing regular physical exercise, which is a similar proportion to that in our reference group of control women; moreover, this was a higher percentage than that of the FM-only group. This might be because, although women with diagnosed CFS subjectively report substantial intolerance to exercise, pharmacological treatment is also much more complicated and less effective than in women with FM alone. Therefore, perhaps the group of women with FM and associated CFS, resort to non-pharmacological therapies as the only alternative, hoping for a considerable improvement in their health and quality of life, despite the difficulty that this entails. In fact, regular physical exercise has been well proven to reduce symptoms in patients with FM, by improving both psychological and physiological aspects such as depression, anxiety, body composition, pain and quality of life [40], and immunophysiological biomarkers that underlie the improvement in quality of life [5,7,8,37]. Similarly, in people with CFS, Dannaway et al. [41] reported positive effects of physical exercise through pathways that reverse physical deconditioning or central sensitization to exertion or activity. However, very little has been described about how exercise influences when both syndromes are concomitant. In the present investigation, we observed that the patients who reported performing HPE also presented improvements in almost all of the mental health and quality of life parameters that were evaluated, without a positive or negative influence of the CFS codiagnosis. Therefore, the HPE reported by the patients, which consisted mostly of physical activities with an important social component, had a positive impact on stress, anxiety, and quality of life and personal control in both experimental groups. Personal control values even became closer to the reference levels of control women, but without differences between the experimental groups. That is, codiagnosis with CFS does not, in general, have an effect on the responses to exercise. A differential effect in FM + CFS and FM-only patients was only observed in depression levels in response to exercise, given that only those without CFS improved. These results are in line with those recently reported by Larun et al. [13] since they found that physical exercise did not improve depression in patients with CFS.

We also decided to evaluate anxiety and fear of COVID-19, since one of the main characteristics of women with FM is the lower capacity to handle stressful situations, and it is important to take into account that altered emotional states caused by additional dysregulation of the limbic system in centrally sensitized patients can intensify symptoms of depression and anxiety [42]. Some studies suggest that symptoms from FM patients who have had COVID-19 got worse, as it can be expected due to their hypersensitivity state [43]. Furthermore, a pilot study by Cankurtaran et al. [44] showed that anxiety and fear of COVID-19 worsened the symptoms and mood of FM patients. Therefore, in our study we had to consider whether women who reported performing HPE during the pandemic might experience less anxiety and fear of COVID-19 than more sedentary patients, and whether the codiagnosis with CFS influenced this matter. Results clearly showed that women who reported performing HPE coped better with the pandemic situation, including fear of contracting the disease, all of this at similar levels to those of healthy women of the same age range. HPE practice encouraged them to go outdoors and continue leading a normal life as far as possible. This finding is supported by Martins et al. [45], who also pointed out that a high proportion of FM patients quit exercising during the COVID-19 pandemic, resulting in an aggravation of the impact of FM during that period.

On the other hand, and paradoxically, although pain is the differentiating, hallmark symptom in FM patients, whereas in CFS patients fatigue is, both experimental groups reported equal pain, but the FM group codiagnosed with CFS reported significantly greater fatigue. However, no differences were observed between the women of the two experimental groups who reported performing physical activity and the sedentary women in any of the indicated parameters including pain. Other studies, using the transcutaneous electrical nerve stimulation (TENS) (FAST) technique in patients with FM, have not observed an improvement in pain either, finding an insignificant relationship between pain and physical activity, although FM patients who performed physical activity did find an improvement in fatigue [46].

Since this research is not an interventional study, it presents the limitations corresponding to an observational study in which it was not possible to control, principally, neither the intensity and type of exercise programs nor the prescribed medication, particularly antidepressant drugs and other possible self-medications. That is why future research focusing whether the health perceived by FM patients corresponds to objective immunoneuroendocrine biomarkers and objective levels of physical activity evaluated through accelerometry will be essential to propose more objective and personalized therapeutic strategies.

## 5. Conclusions

Therefore, we can conclude that codiagnosis with CFS does not negatively affect the already-impaired psychological state and quality of life of FM patients. In addition, HPE has a positive effect on the psychological state and quality of life of patients with FM, without a great influence of the codiagnosis of CFS.

Finally, FM and CFS are still diseases that represent a great challenge to modern medicine, since the etiopathogenesis, diagnosis, and possible therapies are not yet fully clarified. However, we propose that an integrated approach with a realistic individualized therapeutic plan is important in these two conditions and even more so when they occur together, which is in most cases. Further research is crucial, focusing not only on the underlying mechanisms, but also on improvements in exercise-based therapeutic aids that mitigate possible neuroimmunoendocrine dysregulations and also mood disorders; all of this in order to achieve an easier handling of these conditions in the context of psychological care and primary healthcare.

## Figures and Tables

**Figure 1 jcm-11-05735-f001:**
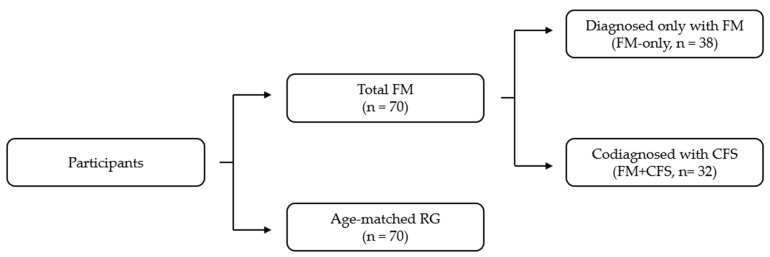
Flow charts of participants in the study. FM = Fibromyalgia, RG = Reference Group, CFS = Chronic Fatigue Syndrome.

**Figure 2 jcm-11-05735-f002:**
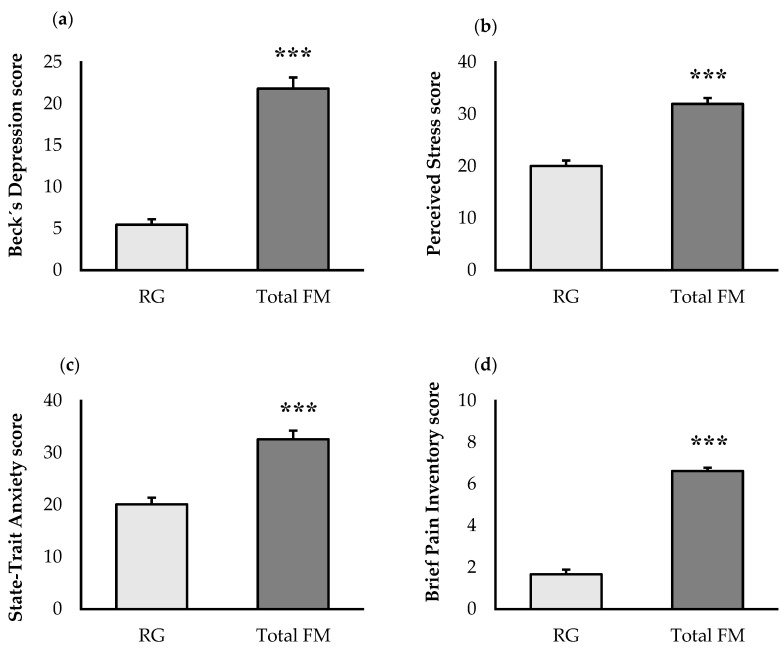
Psychological state and quality of life of FM patients (Total FM, n = 70) compared to an age-matched reference group of “healthy” women (RG, n = 70): perceived levels of (**a**) depression; (**b**) stress; (**c**) anxiety; (**d**) pain; (**e**) fatigue; (**f**) fibromyalgia impact; (**g**) fear of COVID-19; (**h**) coronavirus anxiety. Determinations are expressed by the mean ± SEM of each group. RG: Reference Group, FM: Fibromyalgia. *** *p* < 0.001 with respect to the reference group.

**Figure 3 jcm-11-05735-f003:**
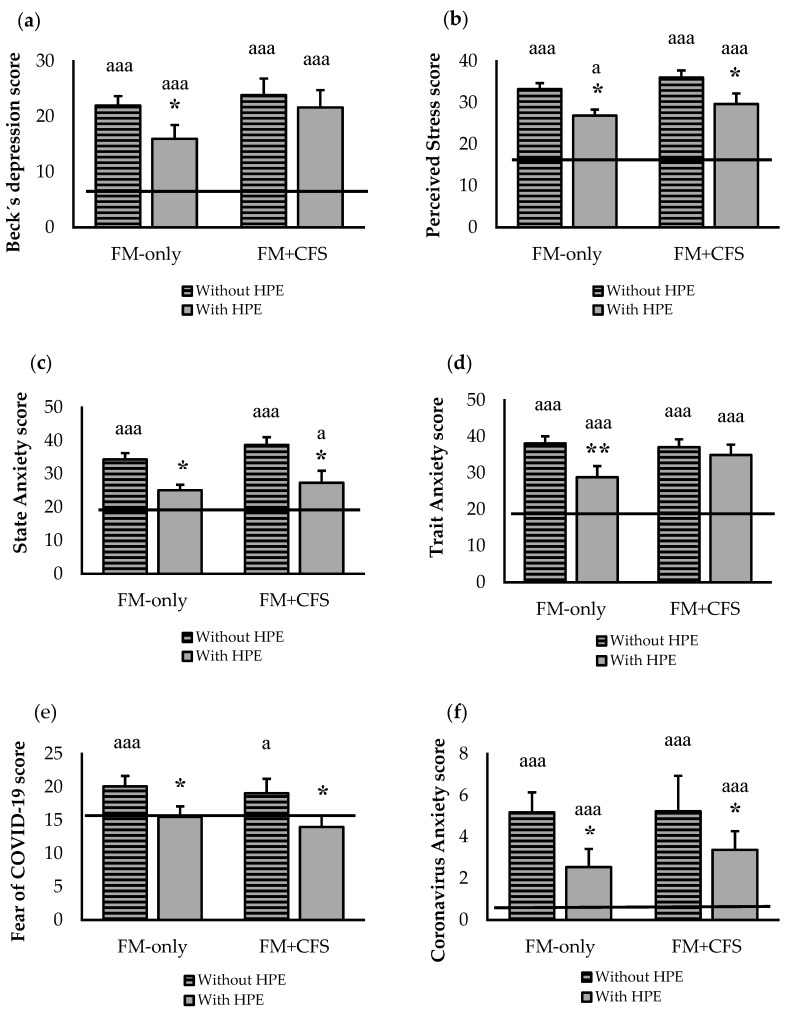
Effect of Habitual Physical Exercise (HPE) in patients diagnosed only with FM (FM-only, n = 38: 27 without HPE and 11 with HPE) and codiagnosed with CFS (FM + CFS, n = 32: 15 without HPE and 17 with HPE), on psychological status and mental health: perceived levels of (**a**) depression; (**b**) stress; (**c**) state anxiety; (**d**) trait anxiety; (**e**) fear of COVID-19 and (**f**) coronavirus anxiety. Horizontal line represents values obtained in the age-matched reference group of “healthy” women without FM. Columns represent the mean ± SEM of each experimental group with or without reported HPE. FM: Fibromyalgia, CFS: Chronic Fatigue Syndrome. * *p* < 0.05, ** *p <* 0.01 with respect to the corresponding group without HPE. ^a^ *p* < 0.05, ^aaa^ *p* < 0.001 with respect to reference group. “Without HPE”: participants non reporting HPE; “With HPE”: participants reporting HPE.

**Figure 4 jcm-11-05735-f004:**
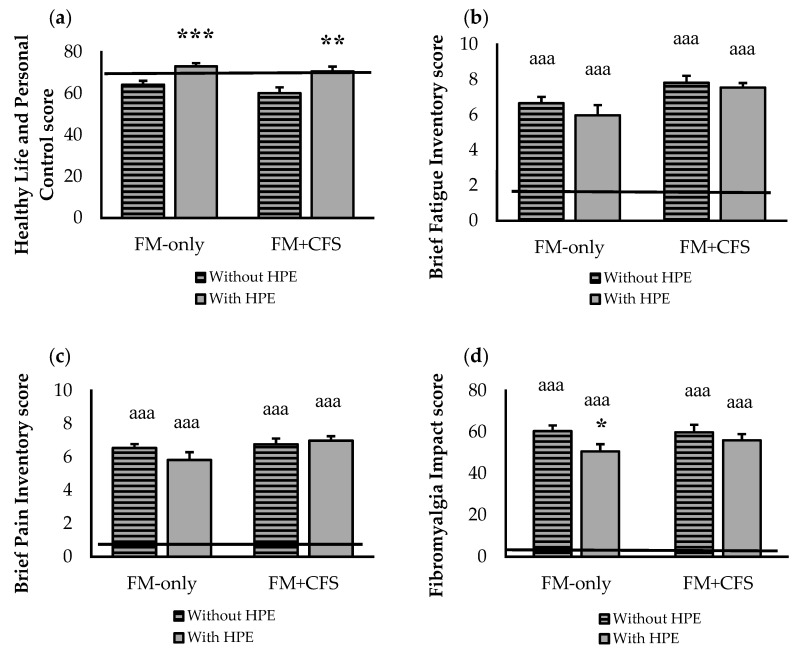
Effect of Habitual Physical Exercise (HPE) in patients diagnosed only FM (FM-only, n = 38: 27 without HPE and 11 with HPE) and codiagnosed with CFS (FM + CFS, n = 32: 15 without HPE and 17 with HPE), on quality of life: perceived levels of (**a**) healthy life and personal control (**b**) fatigue; (**c**) pain and (**d**) fibromyalgia impact. Horizontal line represents values obtained in the age-matched reference group of “healthy” women without FM. Columns represent the mean ± SEM of each experimental group with or without reported HPE. FM: Fibromyalgia, CFS: Chronic Fatigue Syndrome. * *p* < 0.05, ** *p <* 0.01, *** *p* < 0.001 with respect to the corresponding group without HPE. ^aaa^ *p* < 0.001 with respect to reference group. “Without HPE”: participants non reporting HPE; “With HPE”: participants reporting HPE.

**Table 1 jcm-11-05735-t001:** Anthropometric characteristics, employment status, and the most common comorbidities of the participants.

	RG (N = 70)	FM-Only (N = 38)	FM + CFS (N = 32)	Statistical Significance
**Gender (%)**	Women (100%)	Women (100%)	Women (100%)	
**Ethnic group (%)**	Caucasian (100%)	Caucasian (100%)	Caucasian (100%)	
**Age (years)**	53.93 ± 8.27	54.82 ± 8.52	57.31 ± 7.86	*p* > 0.05
**BMI (kg/m^2^)**	25.72 ± 4.75	26.42 ± 5.25	26.82 ± 5.16	*p* > 0.05
**Duration of FM or CFS diagnosed (years)**	___	>2	>2	
***Employment status***:				**Chi-Square (X^2^)** *p* < 0.001 (X^2^ < 0.001)
- **Blue collar workers (%)**	8.6	15.8	12.5	
- **White collar workers (%)**	77.1	28.9 *	18.8 *	
- **Unemployed (%)**	8.6	26.3 *	18.8	
- **Medical leave (%)**	___	5.3	18.6 *	
- **Retired (%)**	5.7	23.7 *	31.3 *	
* **Common comorbidities** *				**Chi-Square (X^2^)**
- **Hypothyroidism (%)**	11.4	28.9	37.5 *	*p* < 0.01 (X^2^ = 0.007)
- **Hypertension (%)**	8.6	13.2	18.8	*p* > 0.05 (X^2^ = 0.332)
- **Osteoarthritis (%)**	___	28.9 *	18.8 *	*p* < 0.001 (X^2^ < 0.001)

Data are expressed as mean ± SEM and as percentage (%). RG: Reference Group, FM: Fibromyalgia, CFS: Chronic Fatigue Syndrome, BMI: Body Mass Index. *****
*p* < 0.05, with respect to reference group (post hoc Z-Test).

**Table 2 jcm-11-05735-t002:** Percentage of subjects reporting Habitual Physical Exercise.

	RG	FM-Only	FM + CFS Chi-Square (X^2^)
**Reporting HPE (%)**	54.29	28.95 *	53.13 *p* < 0.05 (X^2^ = 0.03)
***Type of reported HPE*: **			*p* > 0.05 (X^2^ = 0.32)
- **Walking (%)**	42.1	63.6	58.8
- **Pilates (%)**	15.8	18.2	23.5
- **Yoga (%)**	10.5	9.1	11.8
- **Aquagym (%)**	5.3	9.1	5.9
- **Others (%)**	26.3	___	___

Data are expressed as percentage (%). RG: Reference Group, FM: Fibromyalgia, CFS: Chronic Fatigue Syndrome, HPE: Habitual Physical Exercise, *****
*p <* 0.01 with respect to both reference and FM + CFS groups (post hoc Z-test).

**Table 3 jcm-11-05735-t003:** Influence of habitual physical exercise (HPE) on the values of the HLPCQ items questionnaire in patients with FM, with and without associated CFS.

	RG		FM-Only		FM + CFS	
Items HLPCQ	Without HPE	With HPE	Without HPE	With HPE	Without HPE	With HPE
Dietary Healthy Choices score	14.89 ± 0.52	16.66 ± 0.61 **	16.48 ± 0.70	19.00 ± 0.90 *	15.94 ± 0.94	18.00 ± 0.79 *
Dietary Harm Avoidance score	9.43 ± 0.89	9.05 ± 0.35	10.56 ± 0.43	11.31 ± 0.49	9.75 ± 0.57	11.94 ± 0.58 **
Daily Routine score	24.26 ± 0.25	24.49 ± 0.66	22.89 ± 1.08	24.92 ± 1.11	21.06 ± 1.71	22.88 ± 1.18
Organized Physical Exercise score	3.37 ± 0.46	6.22 ± 0.25 ***	3.56 ± 0.23	5.62 ± 0.58 **	3.75 ± 0.48	5.24 ± 0.49 **
Social and Mental Balance score	12.91 ± 1.61	13.54 ± 0.47	11.44 ± 0.49	11.69 ± 0.68	10.81 ± 0.56	12.24 ± 0.58 *
Total score	65.53 ± 1.61	70.23 ± 1.42 **	64.89 ± 1.74	72.54 ± 1.74 ***	61.31 ± 2.85	70.29 ± 2.41 **

* *p* < 0.05, ** *p* < 0.01, *** *p* < 0.001 with respect to the corresponding group without HPE. Data are expressed as mean ± SEM. RG: Reference Group, FM: Fibromyalgia, CFS: Chronic Fatigue Syndrome, HLPCQ: Healthy Life and Personal Control Questionnaire. “Without HPE”: participants non reporting HPE; “With HPE”: participants reporting HPE.

## Data Availability

The raw data supporting the conclusions of the manuscript will be made available by the authors, without undue reservation, to any qualified researcher.

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
