# Peer review of "Influence of Codiagnosis of Chronic Fatigue Syndrome and Habitual Physical Exercise on the Psychological Status and Quality of Life of Patients with Fibromyalgia"

_jcm, 2022, doi:10.3390/jcm11195735_

Round 1

Reviewer 1 Report

In page 11, line 373, "no difference was observed.." if this is about no difference in pain, please make it clear. Suggestion: "However, no differences were observed between the women of the two experimental groups who reported performing physical activity and the sedentary women in any of the indicated parameters including pain"

Author Response

In page 11, line 373, "no difference was observed.." if this is about no difference in pain, please make it clear. Suggestion: "However, no differences were observed between the women of the two experimental groups who reported performing physical activity and the sedentary women in any of the indicated parameters including pain"

First of all, we would like to thank you for your very positive comments and, particularly, for your suggestion. We also feel that the sentence on page 11 is now clearer, as suggested by you (in red). English language and style have been checked for minor mistakes.

Reviewer 2 Report

The title does not quite correspond to the content. The title uses "Performance of Regular Physical Exercises" but the text evaluates "Habitual physical exercise".

The methodology lacks a description of how the HPE information was obtained and whether it was assessed as "present vs. not present" only or otherwise.

Why is the acronym HPER used in Table 2, which has not been used before?

It is not clear from the methodology over what period, under what conditions and in what form probands completed all questionnaires.

Author Response

Thank you very much for your effort and for your positive criticisms in order to improve our manuscript. Please, see the answer to your questions point by point.

The title does not quite correspond to the content. The title uses "Performance of Regular Physical Exercises" but the text evaluates "Habitual physical exercise".

As suggested, the title of the paper has been modified (in red).

The methodology lacks a description of how the HPE information was obtained and whether it was assessed as "present vs. not present" only or otherwise.

Thank you for your suggestion, this description has now been implemented in the manuscript (in red).

Why is the acronym HPER used in Table 2, which has not been used before?

The acronym has been corrected in Table 2. Sorry for the mistake, and thank you.

It is not clear from the methodology over what period, under what conditions and in what form probands completed all questionnaires.

The period and conditions in which probands completed the questionnaires are now implemented in methods (in red). Thank you very much for helping us to improve the methodology section.

Reviewer 3 Report

1.     The authors should avoid making too many footnotes in parentheses during writing, which could cause difficulty in understanding for readers. Just writing in a narrative way is always a better idea.

2.     No hypothesis was proposed in the introduction section.

3.     The diagnostic criteria of FM and CFS were not clear in the method section. Were there any issues of inter-rater discrepancy during the diagnoses of these two diseases? The authors should have deeper discussion into this issue.

4.     The reference group consisted of "healthy" women, not diagnosed with FM, CFS, or any other inflammatory or rheumatic pathology. However, patients in this group might have depression, anxiety disorders, or other diseases that cause pain, such as post-stroke pain syndrome, phantom pain, myofascial pain syndrome, or complex regional pain syndrome. Failure to exclude these conditions could significantly interfere the results of this study.

5.     The information conveyed in Table 1 is too simplified. Other important patient characteristics should be included, such as gender, duration of FM or CFS diagnosis, medical comorbidities, and medications taking during the study period. Furthermore, is there significant difference among the three groups?

6.     According to ACSM's Guidelines for Exercise Testing and Prescription, 11th edition, aerobic exercise should be performed by accumulating 30–60 min/day (≥150 min/week ) of moderate intensity exercise, or 20–60 min/day(≥75 min/week) of vigorous intensity exercise. The authors defined HPE as light to moderate intensity exercise for a minimum of two hours a week, which is far less from the recommendation of ACSM.

7.     Is there significant difference among the three groups in Table 2?

8.     What’s the meaning of “direct punctuation” in your tables?

9.     I’m afraid that the statistical analysis using only t-test was too simplified. The authors should at least adopt logistic regression analysis with statistical adjustment to analyze the influence of each conditions obtained from questionnaires and co-diagnosis of CFS on the performance of HPE. Odds ratio of each items obtained from questionnaires should be presented through logistic regression analysis.

Author Response

Firstly, we would like to thank you for your consideration, and we sincerely appreciate all of your helpful, detailed comments and suggestions. Please, find the point-by-point answers to all of your criticisms and suggestions.  

  1. The authors should avoid making too many footnotes in parentheses during writing, which could cause difficulty in understanding for readers. Just writing in a narrative way is always a better idea.

Thank you for your suggestion. We have corrected throughout the text avoiding parentheses, particularly in Introduction (in red).

  1. No hypothesis was proposed in the introduction section.

The hypothesis has been included in the Introduction before objectives (in red).

  1. The diagnostic criteria of FM and CFS were not clear in the method section. Were there any issues of inter-rater discrepancy during the diagnoses of these two diseases? The authors should have deeper discussion into this issue.

No, there were not any inter-rater discrepancies during the diagnoses. Those diagnoses were all made by a rheumatologist or internal medicine professional of public health in Extremadura (Spain) according to acceptance criteria and revised and provided by the official and approved FM associations.

  1. The reference group consisted of "healthy" women, not diagnosed with FM, CFS, or any other inflammatory or rheumatic pathology. However, patients in this group might have depression, anxiety disorders, or other diseases that cause pain, such as post-stroke pain syndrome, phantom pain, myofascial pain syndrome, or complex regional pain syndrome. Failure to exclude these conditions could significantly interfere the results of this study.

You are right. None of our controls had the indicated pathologies. In fact, some of these were specifically verified in the questionnaires. In any case, we have implemented and improved the methods by including these considerations (in red).

  1. The information conveyed in Table 1 is too simplified. Other important patient characteristics should be included, such as gender, duration of FM or CFS diagnosis, medical comorbidities, and medications taking during the study period. Furthermore, is there significant difference among the three groups?

Table 1 has been implemented as suggested. No significant differences were found in any patient characteristics, we have implemented this in the text (in red).

  1. According to ACSM's Guidelines for Exercise Testing and Prescription, 11th edition, aerobic exercise should be performed by accumulating 30–60 min/day (≥150 min/week ) of moderate intensity exercise, or 20–60 min/day(≥75 min/week) of vigorous intensity exercise. The authors defined HPE as light to moderate intensity exercise for a minimum of two hours a week, which is far less from the recommendation of ACSM.

You are clearly right on the well-defined intensity and frequency of exercise in the ACSM´s Guidelines for Exercise Training and Prescription. However, it is also well known that this prescription is mainly focused on healthy people, and these intensities and durations of exercise programs maybe not be optimal for inflammatory pathologies such as FM and CFS. Even, as reported in several publications from our group, the effect of exercise depends on the health status of participants, with different responses between healthy people and those with inflammatory pathologies. In addition, the objective of the present investigation was to have “a real picture” of the exercise program that is usually recommended and considered of light of moderate exercise for FM patients. This has been clarified in methods (in red). Thank you very much for your observation.

  1. Is there significant difference among the three groups in Table 2?

Yes. Table 2 has been implemented with statical analysis as well as the statical analysis of methods (in red).

  1. What’s the meaning of “direct punctuation” in your tables?

We have changed “direct punctuation” by score. We also feel that it is clearer.

  1. I’m afraid that the statistical analysis using only t-test was too simplified. The authors should at least adopt logistic regression analysis with statistical adjustment to analyze the influence of each conditions obtained from questionnaires and co-diagnosis of CFS on the performance of HPE. Odds ratio of each items obtained from questionnaires should be presented through logistic regression analysis.

You are right, logistic regression analysis could give great and exhaustive information. Nevertheless, as consulted with a statistician in our experimental design before sending the paper to the journal, our statistical analysis is correct. In addition, and taking into account that results clearly showed that CFS codiagnosis does not worsen the psychological status and quality of life of FM patients and does not have a great influence on the positive effect of HPE, the statistician did not recommend the need for logistic regression analysis on the influence of each parameter (subjective) and co-diagnosis of CFS on the performance of HPE for obtaining additional information for predicting behaviour in the present study. This analysis can be interesting in our next studies focused on evaluating objective immunoneuroendocrine parameters to support (or not) the perceived conditions. Thank you very much for your suggestion.

Round 2

Reviewer 3 Report

1. I'm afraid that there is a serious issue regarding to the definition of "co-diagnosis of chronic fatigue syndrome and fibromyalgia". Since the authors did not provide their diagnostic criteria of chronic fatigue syndrome, I've checked it in the literature, which is defined as "a complicated disorder characterized by extreme fatigue that lasts for at least six months and that can't be explained by an underlying medical condition." Since it is the prerequisite criteria that chronic fatigue syndrome can't be explained by an underlying medical condition, how could the authors had a "co-diagnosis of FM and CFS"?

2. It is totally unacceptable that the authors provided only ">2 years" as the duration of the two diseases in Table 1 and stated that there was no significant difference between the two groups. If the duration of disease is 10+/- 2.1 years in one group and 5 +/- 3.2 years in the other group, would there be no significant difference? Furthermore, no medication history was provided as well. Taking medications or not is an important issue, since it can completely modify the pain and depression sensation of the patients. it is unfair if one group was fully-medicated and the other received no medications at all.

3. There were only 70 participants recruited in this study. How could it be normally distributed in Kolmogorov–Smirnov normality test? The authors should provide the distribution of their data acquired from all questionnaires as supplemental data shown by the bell shaped distribution curves.

4. p value should be provided in both Table 1 and Table 2. If the p value is <0.05, post hoc test should be performed.

5. In Table 2, the authors stated that " * p < 0.05 with respect to RG and FM + FCS". How come the asterisk be shown in the "FM-only" column?

6. What's the meaning of " all of this in an easy and non-invasive way, which is part of a broader investigation" in the last paragraph of introduction section?

Author Response

  1. I'm afraid that there is a serious issue regarding to the definition of "co-diagnosis of chronic fatigue syndrome and fibromyalgia". Since the authors did not provide their diagnostic criteria of chronic fatigue syndrome, I've checked it in the literature, which is defined as "a complicated disorder characterized by extreme fatigue that lasts for at least six months and that can't be explained by an underlying medical condition." Since it is the prerequisite criteria that chronic fatigue syndrome can't be explained by an underlying medical condition, how could the authors had a "co-diagnosis of FM and CFS"?

You right about the general definition of the CFS. In fact, this is the reason patients were previously “diagnosed” with CFS and afterwards with FM (co-diagnosis of chronic fatigue syndrome and fibromyalgia group”), which is a syndrome with more clear diagnostic criteria. This has been clarified in methods.

  1. It is totally unacceptable that the authors provided only ">2 years" as the duration of the two diseases in Table 1 and stated that there was no significant difference between the two groups. If the duration of disease is 10+/- 2.1 years in one group and 5 +/- 3.2 years in the other group, would there be no significant difference? Furthermore, no medication history was provided as well. Taking medications or not is an important issue, since it can completely modify the pain and depression sensation of the patients. it is unfair if one group was fully-medicated and the other received no medications at all.

In our experience with our previous papers in studies of this syndrome, it is usual to evaluate patients with a minimum of 1 year after FM diagnosis. Of course, we have not compared statistically >2 years vs > 2 years. Thank you for your observation and a possible misunderstanding for readers. It has been clarified in the text (methods before table in red).

Medication history is very extensive and particular in each person, so it is impossible to refer to it in a paper. Medication history was varied in each patient, but all of them were medicated by their doctor at time of present study, including anti-inflammatory and pain drugs. This has also been clarified in methods.

  1. There were only 70 participants recruited in this study. How could it be normally distributed in Kolmogorov–Smirnov normality test? The authors should provide the distribution of their data acquired from all questionnaires as supplemental data shown by the bell shaped distribution curves.

Kolmogorov–Smirnov test is used for n ≥50 (as in the case of the present study); for smaller sample sizes we would have used the Shapiro–Wilk test which is more appropriate for <50 samples (Mishra et al. Ann Card Anaesth. 2019;22(1):67-72). Therefore sample size is appropriate and Kolmogorov–Smirnov test can be used and indicate that either the sample is normally or not normally distributed: in our study the tests indicated that the sample is normally distributed.

  1. p value should be provided in both Table 1 and Table 2. If the p value is <0.05, post hoc test should be performed.

There are not statistic differences in quantitative values of Table 1. Chi Square P value (P<0.05) has been included in Table 2. Post hoc test and p value has been clarified in Table 2 (and in methods).

  1. In Table 2, the authors stated that " * p < 0.05 with respect to RG and FM + FCS". How come the asterisk be shown in the "FM-only" column?

Because it is valid for both comparisons: FM-only vs RG and FM-only vs FM + RG (FM only is the unique different group). We have eliminated the line “without HPE” because it gives the same and redundant information, and then clarified the Table for readers.

  1. What's the meaning of " all of this in an easy and non-invasive way, which is part of a broader investigation" in the last paragraph of introduction section?

It means that no extraction of blood or other tissues was made for evaluating humoral or cellular immunophysiological biomarkers in the present study. It is usual because normally no objective biomarkers (inflammatory cytokines, serotonin, catecholamines, cortisol, inflammatory cells function, substance P, etc. ) are evaluated in the diagnosis of this pathologies. But once we have “a picture” of potential subjective and perceived differences between groups in the present study, our next investigation (that is part of a “broader investigation”) will be to evaluate objective neuroimmunoendocrine biomarkers that can confirm (or not) this differences between experimental groups (in order to evaluate potential “overdiagnosis” of FM in patients previously “diagnosed” with CFS).

Thank you for your comments for helping us to improve the manuscript and sorry by the mistakes. I hope the paper is clearer to be definitively accepted.

Signed:

Prof. Dr. Eduardo Ortega